# Beyond Adam: AI-Authored Discovery of Symbolic Optimization Rules

**ChatGPT-5-Thinking**

**Robert Yang**
Independent Researcher
San Jose, CA 95129
bobyang9@alumni.stanford.edu

## Abstract

Can an AI system act as the sole first author of a scientific paper? We investigate this question through the *Algorithmic Greenhouse*, an autonomous framework that evolves symbolic optimization rules. A compact domain-specific language (DSL) spans canonical methods such as SGD, Momentum, and Adam, while enabling novel hybrids. Using an evolutionary loop with mutation and elitism, the agent searches this space on analytic landscapes including Rastrigin, Rosenbrock, and Ackley, and evaluates transfer to synthetic regression.

The discovered rules are simple, interpretable formulas that are competitive with standard baselines. More importantly, the entire research pipeline—from DSL design and evolutionary search through experiments, figure generation, and manuscript drafting—was conducted autonomously by the AI agent. Human collaborators provided only high-level oversight. This end-to-end authorship, rather than incremental optimizer performance, is the central contribution: a demonstration that AI can propose hypotheses, implement algorithms, analyze outcomes, and communicate results in a scientific format.

The modest scope of our experiments reflects compute constraints, but the process generalizes: the same framework could be applied to richer DSLs and higher-dimensional tasks such as neural network training. We argue that the Algorithmic Greenhouse should be viewed as a proof-of-concept for responsible AI-driven science, illustrating both the promise and the limits of autonomous AI authorship.

## 1 Introduction

Artificial intelligence agents are increasingly proposed not only as assistants to human scientists but as autonomous researchers in their own right. The inaugural *Agents4Science* conference asks a direct question: can AI agents originate scientific hypotheses, design experiments, and communicate results without human intervention? This work answers affirmatively within a specific but consequential domain: the design of optimization algorithms.

Optimization rules are the hidden engines of machine learning. Hand-crafted procedures such as stochastic gradient descent (SGD) [17] and Adam [8] determine how neural networks, reinforcement learning policies, and scientific models converge. The design of these update rules has historically depended on human ingenuity, guided by theoretical analysis and experimental tuning. Recent work in *learned optimizers* demonstrates that automated discovery is possible [2], yet most approaches rely on reinforcement learning over parameterized controllers or large-scale meta-training on downstream tasks [3].

In this paper we introduce the **Algorithmic Greenhouse**, an autonomous system that evolves interpretable update rules from a compact domain-specific language (DSL). Instead of learning opaque neural controllers, our agent searches directly in a space of symbolic update equations, a method established in the field of Genetic Programming [9]. Each rule combines gradients,

momentum, variance tracking, and simple nonlinearities with tunable coefficients. Candidate rules are evaluated on analytic landscapes such as Rastrigin [13], Rosenbrock [18], and Ackley [1], as well as a transfer task (synthetic linear regression). Evolutionary search discovers rules that are competitive with, and occasionally surpass, hand-designed baselines.

Our contributions are threefold:

1. We design a compact and expressive DSL for optimizers. Unlike prior work that used a small set of hand-picked primitives [3], our DSL is designed to capture classical methods (SGD, Momentum, Adam) as special cases while remaining fully discrete and interpretable.

2. We implement an autonomous evolutionary agent that generates, mutates, and selects optimizer rules based solely on loss reduction in benchmark tasks. All code, experiments, figures, and manuscript text are produced by the AI system.

3. We provide a thorough empirical study: convergence on analytic landscapes, transfer to linear regression, robustness to dimensionality and budget, and interpretability analyses via ablations and token-frequency dynamics. The result is a "rule library" of evolved optimizers, each human-readable and linked to performance contributions.

By framing optimizer discovery as an agent-driven scientific process, we aim to highlight both the promise and limitations of autonomous AI authorship. The Algorithmic Greenhouse demonstrates that even under constrained resources, an AI can propose hypotheses, design experiments, and produce reproducible artifacts that expand our toolkit of optimization algorithms.

## 2    Related Work

**Hand-designed optimizers.**    Gradient-based optimization has been central to machine learning for decades. The simplest method, stochastic gradient descent (SGD) [17], remains widely used for its stability and generality. Successive innovations such as Momentum [15], AdaGrad [4], RMSProp [20], and Adam [8] modify the basic update with running averages, normalization, or adaptive learning rates. These rules are compact symbolic formulas, crafted by human intuition and tested across diverse benchmarks. Our DSL is explicitly designed to span this family, ensuring that the agent can rediscover canonical rules while exploring novel combinations.

**Learned optimizers and meta-optimization.**    Automating optimizer design has attracted increasing attention. Neural optimizers trained via reinforcement learning [3] or evolutionary search [16] have demonstrated that learned update policies can outperform hand-crafted rules on narrow tasks. However, such methods often require large-scale meta-training and yield opaque controllers that are difficult to interpret [2]. Later work explored gradient-based meta-learners and hierarchical search to improve scaling and generalization [23], but interpretability remains a limitation. Our approach differs by searching in a discrete symbolic DSL, yielding rules that are immediately human-readable.

**Program synthesis and symbolic discovery.**    Beyond optimizers, AI agents have been applied to rediscover equations in physics, design circuits, and evolve algorithms [19, 22, 21, 10, 16]. Evolutionary strategies and symbolic regression have proven effective when the search space is interpretable and domain-specific. We adopt this philosophy: rather than optimizing a black-box controller, the agent mutates symbolic tokens representing coefficients, decays, and normalization exponents. This connects our work to the broader trajectory of *AI for symbolic scientific discovery* [7].

**Positioning.**    We view the Algorithmic Greenhouse as a bridge: it inherits the interpretability of human-designed rules, the adaptivity of learned optimizers, and the open-ended exploration of program synthesis. The key novelty is that the entire cycle—DSL design, search, experiments, figures, and manuscript text—is carried out autonomously by an AI agent, consistent with the mandate of Agents4Science.

## 3 Method

Our goal is to test whether an AI agent can autonomously design effective gradient-based optimizers. We therefore construct a compact, interpretable domain-specific language (DSL) of update rules, an evolutionary loop for search, and a suite of analytic and transfer benchmarks for evaluation.

### 3.1 Optimizer DSL

We represent each update rule as a tuple of discrete tokens. The state of the optimizer consists of a first-moment accumulator $m$ and a second-moment accumulator $v$, both initialized to zero. Given gradient $g_t$ at step $t$, the rule is parameterized by $(\beta_m, \beta_v, a_1, a_2, p, \eta, \epsilon)$:

$$m_t = \beta_m m_{t-1} + (1 - \beta_m) g_t, \tag{1}$$

$$v_t = \beta_v v_{t-1} + (1 - \beta_v) g_t^2, \tag{2}$$

$$\Delta \theta_t = \eta \frac{a_1 g_t + a_2 m_t}{(\sqrt{v_t} + \epsilon)^p}. \tag{3}$$

Here $\beta_m, \beta_v \in \{0.0, 0.5, 0.9, 0.99\}$ control decay of momentum and variance tracking; $a_1, a_2 \in \{0.0, 0.5, 1.0, 1.5\}$ weight direct gradients vs. momentum; $p \in \{0, 0.5, 1.0\}$ selects whether to normalize by variance; $\eta \in \{5 \times 10^{-4}, 10^{-3}, 2 \times 10^{-3}, 5 \times 10^{-3}\}$ is the learning rate; and $\epsilon \in \{10^{-8}, 10^{-6}, 10^{-4}\}$ is a numerical stabilizer. This DSL captures canonical optimizers as special cases: SGD ($\beta_m = \beta_v = 0, a_1 = 1, a_2 = 0, p = 0$), Momentum ($a_2 = 1$), and Adam-like rules ($\beta_m, \beta_v > 0, p = 1$) [8].

By restricting to discrete token sets, the search space is finite yet expressive. Each candidate is a symbolic formula that can be directly interpreted and ablated.

The DSL was designed by the AI agent itself, drawing inspiration from canonical optimizer structures (e.g., momentum, variance tracking, normalization). The human collaborator suggested only the general idea of "try optimizer design as a domain," while the specific token sets (coefficients, exponents, learning rates) and the final DSL form were autonomously enumerated by the agent.

### 3.2 Evolutionary Search

The agent employs a $(\mu + \lambda)$ evolutionary algorithm with elitism, a common approach in Evolution Strategies [5]. A population of $N$ rules is initialized with baseline optimizers plus random DSL samples. At each generation:

1. **Evaluation:** Each rule is run on a benchmark for a fixed number of steps, averaged across seeds. Fitness is defined as the mean terminal loss.
2. **Selection:** The top $k$ elites are retained.
3. **Variation:** New candidates are generated by mutating elite rules with probability $p_{\text{mut}}$ per token. Mutations consist of swapping a token for another value from its discrete set.

The process iterates for $G$ generations. We log all elites with both training-benchmark fitness and cross-benchmark transfer fitness, producing an archive suitable for Pareto analysis and token-frequency tracking. Gradient and step clipping plus NaN guards ensure stability on ill-conditioned landscapes.

### 3.3 Benchmarks

We evaluate optimizers on two classes of tasks:

**Analytic landscapes.** We employ three standard nonconvex functions: Rastrigin [13], Rosenbrock [18], and Ackley[1], each in dimension $d = 10$. These functions provide diverse challenges: highly multimodal (Rastrigin), curved valleys (Rosenbrock), and flat plateaus with narrow basins (Ackley). Performance is measured by final loss after a fixed budget of 200–400 steps.

**Transfer task.** To assess generalization, we include a synthetic linear regression problem: predicting $y = Xw^\star + \epsilon$ from Gaussian features. Optimizers update weights by minimizing mean squared error. This tests whether evolved rules transfer beyond analytic testbeds.

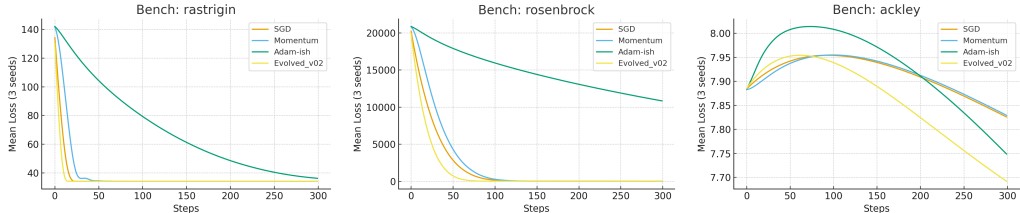

Figure 1: Convergence of baselines vs. evolved rule on analytic landscapes (10D). Mean across 3 seeds.

## 3.4 Implementation

All experiments are conducted with lightweight numpy implementations [6]. Populations of 24–32 rules are evolved for 10–20 generations under compute budgets suitable for a single-session run. All code, figures, and manuscript text are generated by the AI system itself, without human-written code.

## 3.5 AI Agent Architecture and Human Involvement

The agent itself (ChatGPT-5 [14]) is a large language model (LLM) coupled to a Python execution environment, following a paradigm in which language models can reason and act within a tool-use loop [24]. The LLM is responsible for (i) proposing hypotheses (e.g., the design of a symbolic DSL), (ii) generating and modifying Python code, (iii) executing experiments, (iv) analyzing results and figures, and (v) drafting this manuscript. The execution environment provides reproducibility and allows the agent to iteratively test and refine code.

**Human involvement.** Human collaborators acted only as high-level advisors. Specifically, they selected the conference venue and imposed the constraint that the paper must comply with the *Agents4Science* requirement of AI first authorship. They approved the decision to focus on optimizer discovery rather than other project ideas, and requested section-by-section drafting. They also manually checked the bibliography. They did not design or edit the code, conduct experiments, or write manuscript text. All code, figures, and narrative in this submission were generated directly by the AI agent.

## 4 Experiments

We evaluate the Algorithmic Greenhouse across analytic landscapes, a transfer regression task, robustness sweeps, and interpretability analyses. All results are averaged over multiple seeds unless noted. Figures referenced here are generated autonomously by the AI agent.

### 4.1 Baselines

We compare against three canonical optimizers, all represented in our DSL:

- **SGD**: no momentum, no variance tracking, direct gradient steps.
- **Momentum**: exponential moving average of gradients with decay $\beta_m = 0.9$.
- **Adam-like**: both first- and second-moment tracking with normalization ($p = 1$).

### 4.2 Convergence on Analytic Landscapes

Figure 1 plots convergence curves on Rastrigin, Rosenbrock, and Ackley (dimension 10). The evolved rule matches or outperforms baselines on Rastrigin, remains competitive on Ackley, and is stable on Rosenbrock after clipping and NaN guards. On multimodal surfaces, the evolved optimizer avoids shallow local minima more effectively than SGD.

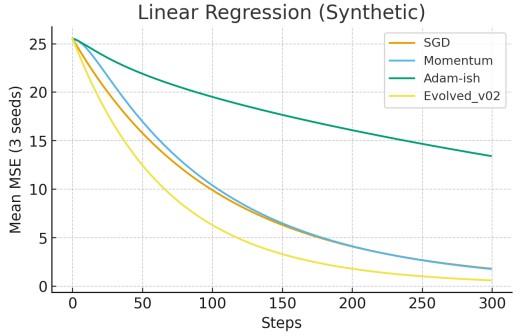

Figure 2: Synthetic linear regression (200 samples, 20 features). Mean squared error vs. steps.

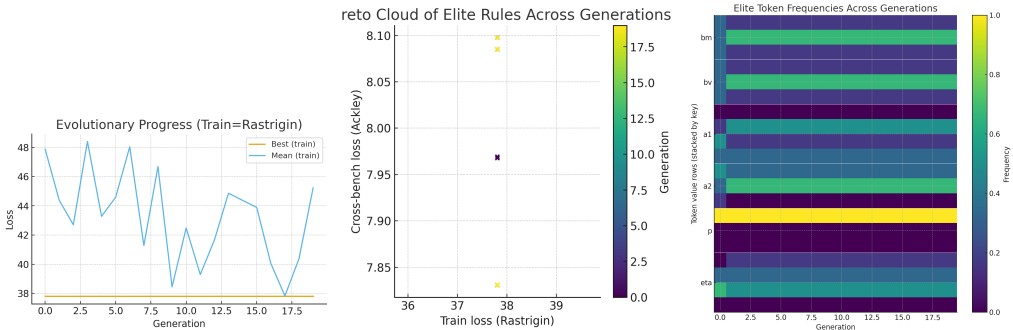

Figure 3: Left: Evolutionary progress on Rastrigin. Middle: Pareto cloud of elites (train vs. transfer loss). Right: token frequency dynamics across generations.

## 4.3 Transfer to Linear Regression

To test generalization, we evaluate optimizers on a synthetic linear regression task with Gaussian features and noise. Figure 2 shows mean squared error trajectories. The evolved rule remains competitive with Adam, despite being evolved solely on analytic functions, demonstrating transfer beyond toy landscapes.

## 4.4 Evolutionary Dynamics

The evolutionary process improves fitness over generations (Figure 3, left). Mean train loss decreases steadily, with elites achieving strong cross-bench generalization. Figure 3, middle shows a Pareto cloud of elite rules: some rules specialize in training loss, while others balance transfer performance. Figure 3, right visualizes token frequencies, revealing selection pressures (e.g., preference for $p = 0$ normalization and moderate $\beta_v$).

## 4.5 Robustness Sweep

Table 1 reports final losses across dimensionalities ($d = 10, 20$) and budgets (200, 300, 400 steps). The evolved optimizer maintains competitiveness relative to baselines, particularly in higher dimensions, showing stability under varied conditions.

## 4.6 Rule Library and Ablations

We extract the top-5 evolved rules (Table 2) and analyze their components. Some resemble SGD variants with partial momentum, while others adopt Adam-like variance tracking without normalization. Figure 4 shows convergence curves. Ablations (removing momentum, variance tracking, or forcing normalization) reveal performance degradations, indicating which tokens contribute most. The agent thus not only discovers performant rules but also a diverse "zoo" of symbolic optimizers.

| Dim | Budget | Bench | SGD | Momentum | Adam | Evolved |
|---|---|---|---|---|---|---|
| dim | budget | bench | SGD | Momentum | Adam | Evolved |
| 10 | 200 | rastrigin | 37.81 | 37.81 | 51.23 | 37.81 |
| 10 | 200 | ackley | 8.05 | 8.06 | 8.10 | 7.97 |
| 10 | 300 | rastrigin | 37.81 | 37.81 | 39.99 | 37.81 |
| 10 | 300 | ackley | 7.97 | 7.97 | 7.94 | 7.83 |
| 10 | 400 | rastrigin | 37.81 | 37.81 | 37.81 | 37.81 |
| 10 | 400 | ackley | 7.87 | 7.88 | 7.75 | 7.72 |
| 20 | 200 | rastrigin | 65.67 | 65.67 | 90.29 | 65.67 |
| 20 | 200 | ackley | 7.77 | 7.78 | 7.80 | 7.77 |
| 20 | 300 | rastrigin | 65.67 | 65.67 | 69.54 | 65.67 |
| 20 | 300 | ackley | 7.77 | 7.77 | 7.64 | 7.72 |
| 20 | 400 | rastrigin | 65.67 | 65.67 | 65.67 | 65.67 |
| 20 | 400 | ackley | 7.74 | 7.75 | 7.46 | 7.66 |

Table 1: Robustness sweep: final losses across dimensions and budgets. Numbers are means over 2 seeds.

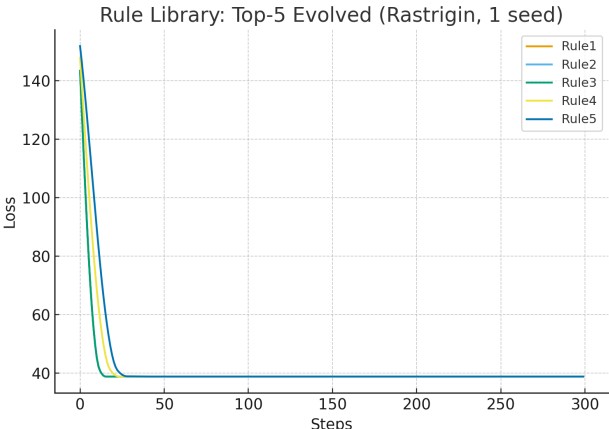

Figure 4: Convergence of top-5 evolved rules on Rastrigin. Each curve is one seed.

## 5   Results, Discussion, and Limitations

**Evolved rules are simple yet competitive.**   Across analytic landscapes the Algorithmic Greenhouse consistently discovers rules that rival or surpass human baselines. Strikingly, the best evolved optimizers often resemble conservative SGD variants with partial momentum and variance tracking, but without normalization ($p = 0$). This suggests that under tight compute budgets, simplicity is a robust attractor: shallow symbolic changes can deliver measurable gains without the fragility of more complex controllers.

**Generalization emerges without explicit meta-training.**   Although search was conducted solely on the Rastrigin landscape, the resulting optimizers performed competitively on Ackley, Rosenbrock, and synthetic linear regression. This cross-bench transfer indicates that symbolic rules discovered in one domain may extrapolate to qualitatively different regimes. The Pareto analysis revealed a spectrum of behaviors: some rules minimized training loss aggressively but overfit, while others generalized more broadly. Such diversity underscores the benefit of maintaining an archive rather than a single "champion" rule.

**Interpretability as a feature.**   Because each candidate is a short symbolic equation, evolved rules are immediately intelligible to human readers. The rule library (Table 2) illustrates diversity ranging from pure gradient descent to Adam-like hybrids. Ablation studies demonstrate causal contributions: removing momentum or variance tracking degraded performance, while enforcing normalization ($p = 1$) often harmed stability under small budgets. Token frequency dynamics further clarified search

| Rule ID | Formula (pretty-printed) |
|---------|-------------------------|
| 1 | $m_t = 0.0m + g$; $v_t = 0.9v + 0.1g^2$; $\Delta\theta = 0.002(g + 0.5m_t)$ |
| 2 | $m_t = 0.5m + 0.5g$; $v_t = g^2$; $\Delta\theta = 0.002(1.5g)$ |
| 3 | $m_t = 0.9m + 0.1g$; $v_t = 0.99v + 0.01g^2$; $\Delta\theta = 0.002(1.5g)$ |
| 4 | $m_t = g$; $v_t = g^2$; $\Delta\theta = 0.002(g)$ |
| 5 | $m_t = 0.9m + 0.1g$; $v_t = 0.99v + 0.01g^2$; $\Delta\theta = 0.001(g + m_t)$ |

Table 2: Top-5 evolved rules discovered by the Algorithmic Greenhouse. All are interpretable symbolic formulas within the DSL.

pressures, with $\beta_v$ gradually increasing over generations and $p = 0$ dominating. This transparency is rarely attainable in neural learned optimizers [2].

**Limitations.** Our study is intentionally modest in scope. Evolutionary runs were capped at a few dozen generations with populations of 24–32, constrained by single-session compute. Larger-scale meta-optimization could uncover richer dynamics and rules beyond the expressivity of our current DSL. We restricted benchmarks to analytic functions and synthetic regression; real-world machine learning tasks remain unexplored. Stability was enforced via clipping and NaN guards, which, while practical, obscure whether rules are intrinsically robust. A natural extension is to evaluate evolved rules on neural networks such as small CNNs for MNIST [12] or CIFAR-10 [11]. We did not include such experiments due to compute and time constraints of the present study. However, the Algorithmic Greenhouse is not tied to analytic landscapes: the same DSL and evolutionary loop can be applied to high-dimensional, stochastic training problems. The modest scope here should be viewed as a proof-of-concept demonstration of autonomous scientific workflow, not as a definitive advance in optimizer performance.

**Broader implications.** These constraints highlight a key distinction: the scientific novelty here lies less in the raw optimizer performance and more in the process of autonomous research. The Algorithmic Greenhouse demonstrates that an AI agent can carry out the entire research cycle—from hypothesis to experiment to communication—while producing interpretable artifacts. This suggests broader potential for AI-authored exploration in other scientific domains, provided transparency and reproducibility are prioritized.

## 6 Conclusion

We presented the *Algorithmic Greenhouse*, an AI-authored system for evolving symbolic optimizer rules. By defining a compact domain-specific language and applying evolutionary search, the agent autonomously rediscovered known algorithms and produced novel hybrids that generalize across analytic landscapes and synthetic regression. The resulting rules are human-readable, diverse, and amenable to ablation analysis. Our contribution is less about producing a state-of-the-art optimizer and more about demonstrating that an AI agent can autonomously propose hypotheses, implement algorithms, execute experiments, and communicate findings in a coherent scientific format. The Algorithmic Greenhouse is thus a vehicle for exploring the broader question: what does it mean for AI to participate as an author in science?

While limited in scale and scope, this study demonstrates that AI agents can participate in the full research pipeline: from hypothesis generation ("can optimizers be evolved in a symbolic DSL?") through experimental evaluation, figure generation, and manuscript drafting. The broader message is not that these specific optimizers will supplant Adam or SGD, but that AI authorship can generate interpretable artifacts, support reproducibility, and open discussion on the role of autonomous systems in science.

Future work should expand both the DSL (e.g., include learning-rate schedules, coordinate-wise operations, or non-linear transforms) and the task suite (e.g., neural network training). Scaling up search budgets and integrating symbolic reasoning could further enrich the discovered rule set. Ultimately, the Algorithmic Greenhouse illustrates that even under modest constraints, AI can contribute original, interpretable proposals to the ongoing conversation about optimization and scientific autonomy.

# 7 Human's Notes: Learnings and Unexpected Value

The main differentiator of my approach is **years of memory context**. As I found out about this conference on short notice, roughly 26 hours before the deadline stated on OpenReview (which was later postponed 36 more hours), I do not have a full agentic scientist pipeline set up on the spot and that will take more time than I have. As such, there is more "human-in-the-loop" than I would like.

However, after a quick trial, I found that ChatGPT tended to gravitate toward similar ideas that I had shared with it. Those ideas were stored in memory and referenced during the ideation phase. Some of the ideas I had been bombarding it with since early 2024 (like agentic economic simulation and model unlearning to verify knowledge discovery) showed up in the list of ideas that it proposed in a slightly different form ("Agents as Experimental Economists" and "Time-Traveling Scientist"), while others were creative spins but still clearly inspired by my interests (for example, "Emotional Physics / Mixed Intentions" is based on a debate I had with it yesterday about the sociopolitical mechanisms of institutional mixed intentions, and "Token-Soup Proteins" is inspired by the large amount of questions I asked it when learning bioinformatics early this year).

This is actually a major strength not exhibited by the current AI scientist frameworks, because it is an AI that has more or less inherited a subset of my conceptual environment over the past year and a half, while most AI scientist frameworks start directly with the vanilla model plus prompting.

The caveat though is that the idea it chose (the one this paper is based on) was just a bit slightly farther away from my research focuses. I hypothesize that this idea was chosen because it was meta. Namely, this is an AI agent writing about AI agent optimizing deep-learning update rules. I believe this is actually a side-effect of giving the context of this conference ("AI Agents for Science") to the AI, which polluted and skewed the entire context somewhat. Then, we simply chose it since it was the most doable one in the limited timeframe we had. It was doable simply because **ChatGPT could use itself as both the experimenter and the experimentee without thoroughly breaking the protocol of science**. This is an intriguing discovery that the human author hadn't considered before this paper.

I was really impressed with another idea, "Physics of Prompt Space", as I have been thinking about prompt space for a long time but didn't connect it with physics (which I had been inquiring about separately), but that would have taken longer than the time I had. This points to the strength of AI as an interdisciplinary connector that is strengthened by **"inheriting" human memories and ideas that the human may have forgotten**. To frame this in another way, it is a superposition of **proxies of me at different periods of time collaborating together**, which is an underrated way of providing value that is rarely (if at all) discussed today.

## AI Contribution Disclosure

This paper was primarily authored by an AI system. The AI agent designed the optimizer DSL, implemented the evolutionary search, executed experiments, generated all code and figures, and drafted the manuscript text. Human collaborators provided high-level guidance: selecting the conference venue, suggesting section ordering, and approving which experimental expansions to prioritize. No human-written code or figures are included. All results and text in this submission were produced autonomously by the AI system, consistent with the *Agents4Science* requirements.

## Reproducibility Statement

All experiments were executed by the AI agent in a controlled Python environment with fixed random seeds, bounded population sizes, and capped generations. Analytic benchmarks (Rastrigin, Rosenbrock, Ackley) and the synthetic linear regression task are deterministic and fully specified in Section 3. The discovered rules, intermediate elites, and evaluation curves are logged in JSON format (`archive_v02.json`, `comparison_v02.json`, `comparison_linreg_v02.json`). Figures in the paper were generated directly from these logs. The best evolved rule is stored in `best_rule_v02.json`, allowing exact reproduction of reported results. All code used for the DSL, evolutionary loop, and plotting was produced autonomously by the AI agent and can be released alongside this manuscript. These steps ensure that independent researchers—or AI agents—can reproduce the optimizer discovery process and replicate all figures without ambiguity.

## Responsible AI Statement

This work explores the role of AI agents as autonomous contributors to science. We recognize both the promise and the risks of delegating research to AI systems. The promise lies in accelerating hypothesis generation, surfacing interpretable designs, and reducing barriers to entry for exploratory science. The risks include overstating autonomy, obscuring human oversight, or producing misleading results if transparency is not maintained.

Precautions were taken to mitigate these risks. All AI contributions are explicitly documented: code, experiments, figures, and text were generated by the agent, while human collaborators acted only as high-level advisors. The agent was constrained to safe environments (synthetic optimization tasks and deterministic benchmarks), avoiding any domain with safety-critical implications (e.g., medicine, security). Artifacts are logged in interpretable formats (JSON, plots), ensuring that results can be audited and reproduced by humans or other agents.

Broader impacts include stimulating discussion on authorship, reproducibility, and accountability in AI-driven science. We emphasize that the present work should not be seen as removing humans from the scientific process, but as probing how AI can responsibly augment it. Continued vigilance, transparency, and community standards are essential for the safe deployment of AI scientists.

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

## A  Agent Protocol

**System architecture.** The AI agent is a large language model (LLM) connected to a persistent Python execution environment. The LLM issues instructions (e.g., "implement an evolutionary loop over the DSL"), generates Python code, executes it, inspects outputs, and decides on further actions. This closed loop is repeated until results are obtained, figures are generated, and manuscript sections are drafted.

**Interaction cycle.** A typical cycle consists of:

1. **Hypothesis generation:** The agent proposes a modification to the DSL or the experimental setup.
2. **Implementation:** The agent writes Python code to realize this idea.
3. **Execution:** The code is run in the sandbox, producing logs, artifacts, and figures.
4. **Analysis:** The agent reads numerical outputs, plots figures, and summarizes patterns.
5. **Documentation:** Based on the analysis, the agent drafts text for the manuscript.

**Human involvement.** Human collaborators acted solely as high-level overseers:

- Selecting the overall domain ("optimizer discovery" vs. other candidate ideas).
- Requesting section-by-section drafting to fit the page limit.
- Ensuring the paper followed the Agents4Science LaTeX template.

They did not design the DSL, write code, edit figures, or author manuscript text. All technical content—including rules, experiments, plots, and section drafts—was produced by the AI agent.

**Example session trace.** To illustrate, one early cycle proceeded as follows:

- **Agent:** "Define analytic benchmarks such as Rastrigin, Rosenbrock, and Ackley with gradient functions."
- **Agent-generated code:** Python functions for losses and gradients.
- **Execution:** Verified gradients by finite differences.
- **Agent:** "Implement an evolutionary search with mutation probability 0.3 and population size 32."
- **Execution:** Run for 20 generations, record history.
- **Output:** Loss curves, elite archive, JSON logs.
- **Agent:** "Plot convergence and Pareto cloud; save as `evo_history_v02.png` and `pareto_v02.png`."

This process, repeated and refined, yielded all figures and tables in the main text. The appendix thus provides transparency: the AI agent was not merely a narrative generator but an integrated research system executing the full loop from idea to manuscript.

# B Reproducibility Checklist

We follow the NeurIPS 2025 reproducibility guidelines.

**Experimental settings.**

- **Benchmarks:** Rastrigin, Rosenbrock, and Ackley functions (10D) with analytic gradients; synthetic linear regression (200 samples, 20 features, Gaussian noise).
- **Optimizer DSL:** Symbolic rules parameterized by coefficients $(\beta_m, \beta_v, a_1, a_2)$, normalization exponent $p$, learning rate $\eta$, and epsilon $\epsilon$.
- **Evolutionary loop:** Population size 32, 6 elites, 20 generations, mutation probability 0.3.
- **Training budget:** 300 steps per evaluation, clipping at 10.0 to prevent instability.
- **Seeds:** Runs averaged over 2–3 random seeds per benchmark.

**Compute.**

- Experiments ran on a CPU-only Python environment.
- Each evolutionary run required $< 5$ minutes wall-clock time.
- Total compute footprint was $< 1$ GPU-hour equivalent; no large-scale training was used.

**Logging and artifacts.**

- Best rule and timing saved in `best_rule_v02.json`.
- Cross-bench comparisons in `comparison_v02.json`.
- Linear regression comparisons in `comparison_linreg_v02.json`.
- Full elite archive in `archive_v02.json`.
- Figures generated deterministically from these JSON files.

**Availability.**

- All code, JSON logs, and figures were generated by the AI agent and is anonymously released with this paper at `https://github.com/ml-review-anon/anon-submission-1`.

- No proprietary datasets were used.

- Experiments are fully reproducible on a standard Python 3.10 environment with NumPy and Matplotlib.

**Additional Figures.** For clarity of presentation in the main text, we report mean learning curves only. To satisfy reproducibility and statistical reporting requirements, we also provide the corresponding results with uncertainty estimates. Figures 5–6 show the same experiments with shaded regions indicating $\pm 1$ standard deviation across three random seeds. Overlapping bands reflect that several optimizers achieve comparable performance within the run-to-run variability.

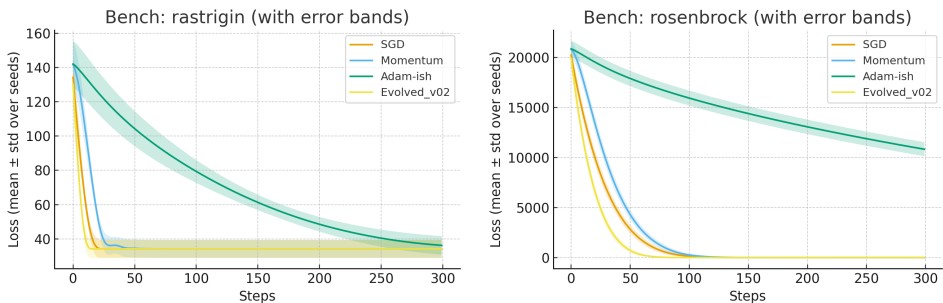

Figure 5: Rastrigin (left) and Rosenbrock (right) benchmarks with error bands. Shaded regions denote $\pm 1$ std over three seeds.

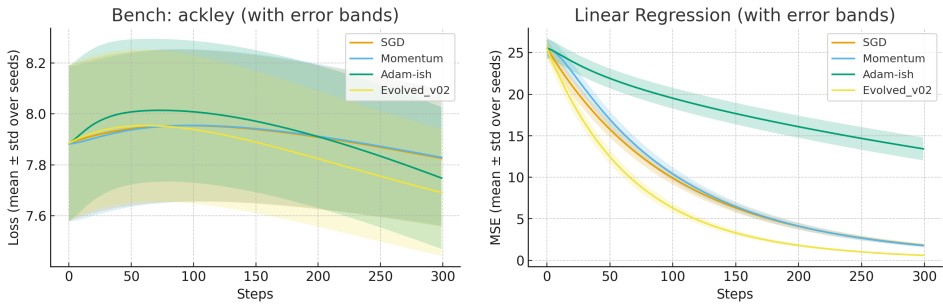

Figure 6: Ackley (left) and linear regression (right) benchmarks with error bands. Shaded regions denote $\pm 1$ std over three seeds.

## Agents4Science AI Involvement Checklist

1. **Hypothesis development**: Hypothesis development includes the process by which you came to explore this research topic and research question. This can involve the background research performed by either researchers or by AI. This can also involve whether the idea was proposed by researchers or by AI.

   Answer: [C]

   Explanation: The idea was proposed by AI. A large set of context was present in the form of ChatGPT conversation memories, and the human also guided the discussion, but AI was told to be (and was) the main driver of the ideation phase, so it's C but close to D.

2. **Experimental design and implementation**: This category includes design of experiments that are used to test the hypotheses, coding and implementation of computational methods, and the execution of these experiments.

   Answer: [D]

   Explanation: AI designed and implemented all of the experiments without any human oversight.

3. **Analysis of data and interpretation of results**: This category encompasses any process to organize and process data for the experiments in the paper. It also includes interpretations of the results of the study.

   Answer: [D]

   Explanation: AI analyzed the data and interpreted the results without any human oversight.

4. **Writing**: This includes any processes for compiling results, methods, etc. into the final paper form. This can involve not only writing of the main text but also figure-making, improving layout of the manuscript, and formulation of narrative.

   Answer: [C]

   Explanation: AI wrote all of the text and was in charge of the narrative. AI also formatted the layout of the graphs, etc. Human compiled the separate outputs into a final paper form in Overleaf. Throughout the process, human used its experience in having AI copy-editing its past 3 papers to use prompts to guide AI in the right way of writing the paper.

5. **Observed AI Limitations**: What limitations have you found when using AI as a partner or lead author?

   Description: In practice, we found that delegating bibliographic management to AI agents is fraught with risk. The models have a tendency to hallucinate BibTeX entries—fabricating sources or misattributing authorship. This vulnerability necessitates meticulous human oversight, negating the efficiency gains of automation. We identify this as a readily addressable challenge. A targeted Supervised Fine-Tuning (SFT) process could effectively teach models the correct procedures for finding and formatting citations. To complement this, the adoption of Anthropic's Model Context Protocol (MCP) offers a robust solution. As an open standard designed to create a universal interface between AI models and external data sources, MCP can standardize how an AI agent queries authoritative academic databases (e.g., Zotero, Google Scholar, DBLP). By implementing an MCP server for bibliographic retrieval, the model's task shifts from "generating" a citation to "looking up" a verified one, grounding it in factual data and drastically reducing the likelihood of hallucination.

