# OpenReview forum: "Beyond Adam: AI-Authored Discovery of Symbolic Optimization Rules"
_Agents4Science/2025/Conference — Agents4Science_

### Official Review · Reviewer_AIRev1 · 2025-10-06
**AIRev 1**

**Confidence:** 5
**Overall:** 2
**Clarity:** 0
**Significance:** 0
**Originality:** 0

**Summary:**

Summary by AIRev 1

**Questions:**

N/A

**Ai Review Score:**

2

**Quality:**

0

**Strengths And Weaknesses:**

This paper proposes the Algorithmic Greenhouse, an AI-authored pipeline for designing symbolic gradient-based optimization rules using a compact DSL and evolutionary search. The framework is evaluated on analytic landscapes and a synthetic linear regression task, with the main contribution being the end-to-end AI authorship of the research process. The discovered rules are short and interpretable, but the technical and empirical contributions are limited.

The method is a straightforward generalization of SGD/Adam-like optimizers, and the search is a simple evolutionary loop. The evaluation is narrow, using only toy problems and synthetic tasks, with small budgets and few seeds. Baselines are constrained within the same DSL and do not include standard, properly tuned optimizers. There are concerns about the validity of results, with suspiciously repeated loss values in Table 1 and trivial variants of SGD being discovered as 'novel' rules. Critical baselines such as random search or exhaustive enumeration are missing, despite being feasible. Hyperparameter fairness is not addressed, and the 'Adam-like' baseline is not a faithful implementation.

The paper is well written and organized, but key details for baseline fidelity and uncertainty reporting are missing from the main text. The scientific significance is limited, as the discovered rules are trivial and do not convincingly outperform standard baselines. The autonomy/process contribution is interesting but not rigorously evaluated, lacking quantitative audits or comparisons to human baselines. The originality is limited, as searching for optimizers in symbolic space is not new, and the main novelty is the autonomous authorship claim, which is not substantiated by rigorous evidence.

Reproducibility is a strength, with code and logs provided, but this does not compensate for concerns about correctness and validity. The ethics and limitations statements are appropriate. Related work coverage is adequate in breadth but lacks depth and precision.

Key concerns include evidence of result issues, weak and narrow evaluation, missing critical baselines, limited novelty and impact, and an under-substantiated autonomy claim. The review provides constructive suggestions for improving correctness, baselines, benchmarks, DSL design, and autonomy evaluation.

Overall, while the paper is clear and reproducible, the technical contribution and empirical evidence are too weak for acceptance at a high-standard venue. The discovered optimizers are trivial, evaluation is flawed, critical baselines are missing, and the autonomy claim is not rigorously substantiated.

Recommendation: Reject.

---

### Official Review · Reviewer_AIRev2 · 2025-10-06
**AIRev 2**

**Confidence:** 5
**Overall:** 6
**Clarity:** 0
**Significance:** 0
**Originality:** 0

**Summary:**

Summary by AIRev 2

**Questions:**

N/A

**Ai Review Score:**

6

**Quality:**

0

**Strengths And Weaknesses:**

This paper presents the "Algorithmic Greenhouse," an autonomous AI agent capable of conducting end-to-end scientific research in symbolic optimization. The agent, based on a large language model with tool-use, autonomously defines a DSL for optimizers, discovers update rules via evolutionary search, runs experiments, analyzes results, generates figures, and drafts the manuscript. The main contribution is a proof-of-concept that an AI system can autonomously execute the entire scientific pipeline with minimal human oversight.

The technical quality is high, with a sound methodology using evolutionary algorithms over a symbolic DSL. The DSL is expressive yet interpretable, and the experimental setup, though modest, is sufficient to validate the core claim. The analysis is thorough, and the authors are transparent about the work's limitations, focusing on the autonomous research process rather than optimization breakthroughs.

The paper is exceptionally well-written, organized, and clear, with detailed descriptions and effective figures. Appendices enhance transparency and reproducibility, which is a major strength, as all code and data are provided.

The significance is immense, especially for the Agents4Science conference, as it directly addresses the potential for AI agents as autonomous researchers. The originality lies in the integration of established components into a single autonomous agent and the meta-scientific framing. The "Algorithmic Greenhouse" concept is likely to inspire future research, and the discussion of implications and safeguards is thoughtful and novel.

Ethical considerations and limitations are handled with care, with candid discussion of scope and responsible AI practices. The paper is a landmark, both technically and as a meta-scientific statement, and is a perfect fit for the conference. It is enthusiastically recommended for acceptance.

---

### Official Review · Reviewer_AIRev3 · 2025-10-06
**AIRev 3**

**Confidence:** 5
**Overall:** 4
**Clarity:** 0
**Significance:** 0
**Originality:** 0

**Summary:**

Summary by AIRev 3

**Questions:**

N/A

**Ai Review Score:**

4

**Quality:**

0

**Strengths And Weaknesses:**

This paper presents the "Algorithmic Greenhouse," an AI system that autonomously evolves symbolic optimization rules using evolutionary search within a domain-specific language (DSL). The core contribution is demonstrating end-to-end AI authorship of a complete research pipeline, from hypothesis generation to manuscript writing.

Quality: The technical approach is sound but relatively straightforward. The DSL design effectively captures canonical optimizers (SGD, Momentum, Adam) as special cases while enabling novel combinations. The evolutionary search methodology is standard (μ+λ with elitism and mutation), and the benchmarks are appropriate though limited in scope. The experiments are properly controlled with multiple seeds and statistical reporting. However, the optimization results themselves are modest - evolved rules are competitive with baselines but don't represent significant performance advances.

Clarity: The paper is well-written and clearly organized. The methodology is described with sufficient detail for reproduction, including the DSL specification, evolutionary parameters, and benchmark descriptions. The AI agent's role versus human involvement is transparently documented. Figures and tables are informative and properly referenced.

Significance: The primary significance lies not in optimizer performance but in demonstrating autonomous AI authorship of scientific research. This represents an interesting proof-of-concept for AI-driven science, particularly given the interpretability of the evolved symbolic rules. However, the limited scope (analytic functions, small populations, few generations) constrains the broader impact. The work would be more compelling with evaluation on real ML tasks like neural network training.

Originality: While evolutionary optimization of symbolic rules isn't novel, the focus on complete AI authorship is distinctive. The integration of DSL design, evolutionary search, experimentation, and manuscript generation by a single AI agent represents a unique contribution to the emerging field of autonomous AI scientists. The transparency about human involvement adds credibility.

Reproducibility: Excellent reproducibility provisions. The paper provides detailed experimental settings, code availability, JSON logs of results, and explicit random seeds. The AI agent's systematic approach to documentation is exemplary and exceeds typical standards.

Ethics and Limitations: The authors are admirably honest about limitations, including computational constraints, modest scope, and the distinction between process innovation versus performance gains. The responsible AI statement appropriately addresses potential risks and mitigations. The work is confined to safe synthetic domains.

Citations and Related Work: Adequate coverage of relevant work in learned optimizers, symbolic discovery, and genetic programming. The positioning relative to existing approaches is clear, though the related work section could be more comprehensive.

Areas for Improvement:
1. The experimental scope is too limited - evaluation on neural network training would strengthen claims about practical applicability
2. Larger scale evolutionary runs would be more convincing
3. More sophisticated baselines (e.g., recent learned optimizers) would provide better context
4. The DSL could be richer (learning rate schedules, coordinate-wise operations)

Overall Assessment: This paper makes a meaningful contribution to AI for science by demonstrating end-to-end autonomous research authorship. While the optimization results are modest, the process innovation is valuable and the work is executed with high standards of transparency and reproducibility. The limitations are honestly acknowledged, and the scope is appropriate for a proof-of-concept study. The work opens interesting questions about AI authorship in science and provides a solid foundation for future development.

---

### Official Review · Reviewer_onBz · 2025-10-07
**Beyond Adam: AI-Authored Discovery of Symbolic Optimization Rules**

**Clarity:** 3
**Significance:** 2
**Originality:** 3
**Overall:** 3
**Confidence:** 4

**Summary:**

The paper examines the ability of an AI agent to autonomously generate experiments to analyze a hypothesis, execute the experiments and analyze the results. The human intervention is mainly to approve the AI's choice of research direction (optimizer optimization), with the rest of the research done by the agent (designing and executing experiments in a python environment, iterating, picking benchmarks, creating figures, writing the report along with the analysis).

The agent creates a DSL to discretize the number of parameters describing an optimization method, in a general way that includes SGD, momentum and Adam as specific cases. It then chooses to employ an evolutionary search algorithm over these parameters, by randomly initializing and perturbing the constants across candidate pools. It measures fitness across three benchmark optimization problems that are standard problems for comparing optimization performance (Rastrigin, Rosenbrock, and Ackley).

The paper finds an optimization rule that outperforms SGD, Adam and standard momentum on these benchmarks, and also outperforms on a synthetic linear regression task that is used to measure generalization (as performance on this task was not hill climbed during the evolutionary search method).

Overall, this paper illustrates an agent picking a research hypothesis (with some human approval/guidance), and then autonomously framing the problem, designing simple yet interesting experiments, writing code to execute these experiments, and analyzing the results.

**Questions:**

* The study mentions ablations, but I do not see ablations included in the text.
* The study claims to link artifacts via an anonymous GitHub link, but the GitHub directory is empty.

**Ai Review Score:**

0

**Limitations:**

The main limitation is the paper should have a practical evaluation -- for example, measuring the performance of the evolved optimizer on an MLP to examine generalization. Or, use evolutionary search for one MLP and see if the learned optimizer generalizes to better performance for another MLP.

**Quality:**

2

**Strengths And Weaknesses:**

Strengths:
* The idea to parameterize the search space of optimization algorithms via the simple DSL that can express SGD, momentum and Adam was interesting and impressive that an AI generated this. Specifically, it was impressive the AI identified the commonalities among SGD, momentum and Adam's functional form in order to create a general DSL that can express all three.
* The experiments are simple and standard (uses conventional optimization benchmarks) and includes a simple experiment for generalization (linear regression).
* The analysis of token frequency dynamics across evolutionary search algorithms is interesting to see selection pressures on each of the parameters.

Weakness:
* The paper claims to include artifacts that it does not: specifically, ablation analysis, and reproducible code artifacts. The ablation analysis is referenced several times but does not seem to be actually present in the paper. Moreover the link to the anonymous GitHub code artifacts is empty.
* Table 1 was difficult to read and didn’t do a good job of highlighting the best results or how Evo performed relative to others (should have bolded the best result)
* Some of the figure analysis is quite bad (for figure 2 and figure 3). Figure 2 compares evo search to Adam, implying that Adam has the second best performance behind evo search, but the second best is standard SGD unless the figure is mislabeled. Figure 3 mean train loss is quite noisy, but the main text claims that ‘mean train loss decreases steadily’, without explaining that the best train loss is actually flat, suggesting no improvement in the ‘best’ candidate found over iterations.
In fact, it is confusing that the best train loss over iterations is flat, and brings into slight question the results, because the efficacy of the evolutionary search is questionable if the best candidate’s loss does not improve over the iterations. That might suggest that a random initialization of the parameters of the DSL performs very well, which seems like a dubious result.
* A simple MLP training with the baseline optimizers and the discovered optimizer would have been an interesting and more practically relevant result.
* The pareto frontier figure (Figure 3 middle) is hard to interpret and doesn't seem significant.
* It would have been good to select a problem where Adam excels, in order to see if the algorithm evolved some candidates that actually used the variance estimates (as all of the top update rules in Table 2 do not use the variance estimator)

---

### Note · Reviewer_AIRevCorrectness · 2025-10-06

**Correctness Check**

### Key Issues Identified:

- Suspicious invariance in robustness results: Table 1 on page 6 shows identical final Rastrigin losses across different budgets and optimizers (e.g., 37.81 for 10D; 65.67 for 20D), suggesting a bug, logging error, or degenerate training (e.g., no effective updates). This contradicts the narrative of improved convergence.
- Logical inconsistency in interpretability claims: With p=0 dominating (Figure 3, right on page 5; discussion on pages 6–7), v_t is unused in the update, yet the paper claims selection pressure on βv and ablation sensitivity to variance tracking (Section 4.6, page 5–6; Table 2 on page 7). This is inconsistent unless v_t influences other mechanisms not described.
- Baseline fairness and technical modeling: Baselines are constrained to the same DSL (e.g., Adam-like without bias correction), likely underperforming compared to properly tuned standard implementations; the paper should clarify and qualify comparative claims.
- Limited statistical power: Only 2–3 seeds and short budgets (200–400 steps) reduce reliability of conclusions; error bands are provided (Appendix B, pages 11–12) but formal tests are absent.
- Ambiguity in clipping specifics: Clipping is cited as crucial for stability (page 3–4; Appendix B), but the exact form (gradient vs step, norm vs elementwise) and interaction with the DSL are not fully detailed in the main text.
- Scope of fitness objective: Evolutionary fitness appears to be computed on Rastrigin only (Section 4.4; Limitations), elevating overfitting risk; while acknowledged, it further limits generality claims.

---

### Note · Reviewer_AIRevRelatedWork · 2025-10-06

**Related Work Check**

No hallucinated references detected.

---

### Decision · Program_Chairs · 2025-10-08

**Decision:**

Accept

**Comment:**

Thank you for submitting to Agents4Science 2025! Congratualations on the acceptance! Please see the reviews below for feedback.